# Targeting AKT/mTOR in Oral Cancer: Mechanisms and Advances in Clinical Trials

**DOI:** 10.3390/ijms21093285

**Published:** 2020-05-06

**Authors:** Choudhary Harsha, Kishore Banik, Hui Li Ang, Sosmitha Girisa, Rajesh Vikkurthi, Dey Parama, Varsha Rana, Bano Shabnam, Elina Khatoon, Alan Prem Kumar, Ajaikumar B. Kunnumakkara

**Affiliations:** 1Cancer Biology Laboratory and DBT-AIST International Center for Translational and Environmental Research (DAICENTER), Department of Biosciences and Bioengineering, Indian Institute of Technology Guwahati, Assam 781039, India; harsha.choudhary@iitg.ac.in (C.H.); kishore.banik@iitg.ac.in (K.B.); sosmi176106101@iitg.ac.in (S.G.); rajes174106009@alumni.iitg.ac.in (R.V.); deyparama@iitg.ac.in (D.P.); varsharana@iitg.ac.in (V.R.); bano176106104@iitg.ac.in (B.S.); elinakhatoon@gmail.com (E.K.); 2Department of Pharmacology, Yong Loo Lin School of Medicine, National University of Singapore, Singapore 117600, Singapore; e0336095@u.nus.edu; 3Cancer Science Institute of Singapore, National University of Singapore, Singapore 117599, Singapore

**Keywords:** Akt, mTOR, oral cancer, inhibitors, treatment, pathway

## Abstract

Oral cancer (OC) is a devastating disease that takes the lives of lots of people globally every year. The current spectrum of treatment modalities does not meet the needs of the patients. The disease heterogeneity demands personalized medicine or targeted therapies. Therefore, there is an urgent need to identify potential targets for the treatment of OC. Abundant evidence has suggested that the components of the protein kinase B (AKT)/ mammalian target of rapamycin (mTOR) pathway are intrinsic factors for carcinogenesis. The AKT protein is central to the proliferation and survival of normal and cancer cells, and its downstream protein, mTOR, also plays an indispensable role in the cellular processes. The wide involvement of the AKT/mTOR pathway has been noted in oral squamous cell carcinoma (OSCC). This axis significantly regulates the various hallmarks of cancer, like proliferation, survival, angiogenesis, invasion, metastasis, autophagy, and epithelial-to-mesenchymal transition (EMT). Activated AKT/mTOR signaling is also associated with circadian signaling, chemoresistance and radio-resistance in OC cells. Several miRNAs, circRNAs and lncRNAs also modulate this pathway. The association of this axis with the process of tumorigenesis has culminated in the identification of its specific inhibitors for the prevention and treatment of OC. In this review, we discussed the significance of AKT/mTOR signaling in OC and its potential as a therapeutic target for the management of OC. This article also provided an update on several AKT/mTOR inhibitors that emerged as promising candidates for therapeutic interventions against OC/head and neck cancer (HNC) in clinical studies.

## 1. Introduction

Oral cancer (OC), one of the most common forms of head and neck cancer (HNC), comprises of cancers of the lip, tongue, gum, palate, mouth, the floor of the mouth, gingiva and other parts of the oral cavity. The most common variant of OC is oral squamous cell carcinoma (OSCC), which comprises ~91% of all OC cases [1]. OC is a global health concern as it is one of the highest mortality causing cancers in the world [2]. As per reports from Globocan in 2018, lip and oral cavity cancer account for 354,864 new cases and 177,384 death cases worldwide annually [3]. The incidence of this cancer is significantly high in Southern Asia and the Pacific islands, among which countries like India and Sri Lanka hold the highest position in regards to the male death toll due to OC [3]. 

The current strategies for the treatment of OC include surgical resection combined with or without radiotherapy or adjuvant chemotherapy. Immunotherapy is another promising approach, but it has not remarkably yielded yet, in the case of OC. Advancements in surgical techniques and the associated therapies for the treatment of OC have undoubtedly increased the five-year survival rate in patients; however, the scenario is still terrifying in particular regions of Asia where the traditional habits of tobacco chewing, smoking, alcohol drinking and betel quid chewing are highly predominant and tertiary healthcare is limited or unavailable for a majority of the population [4,5,6,7,8,9]. These countries demand organized prevention and more access to early detection and treatment of OC.

The high incidence of OC in low resource countries is due to several customary habits such as alcohol consumption, smoking, tobacco chewing and areca nut chewing [10]. OC may also occur due to human papillomavirus (HPV) infection, poor dental care, poor hygiene and consumption of unhealthy food. Additionally, gene polymorphisms and other genetic aberrations also contribute to the pathogenesis of OC. The cancer genome atlas (TCGA) study has revealed that a greater number of the head and neck squamous cell carcinoma (HNSCC) cases have shown an alteration in the protein kinase B (AKT)/ mammalian target of rapamycin (mTOR) pathway [11]. Moreover, compounds like nicotine present in the major risk factor tobacco and the synthetic carcinogen, 4-NQO, have also been reported to induce the activation of the AKT/mTOR pathway [12]. Thus, this pathway is remarkably associated with the development and progression of OC.

In this era of precision medicine, progress has been made in the development of personalized, targeted therapies that might block an individual pathway or a combination of pathways and rescue cancer development or progression. The current commercially available targeted regimen for OC patients includes the epidermal growth factor receptor (EGFR) inhibitor and cetuximab (Cet), also known as Erbitux®. Despite promising results in preclinical studies, resistance to EGFR therapy is a noteworthy limitation for the use of this drug in clinical settings. The increasing incidence of OC thus requires more efficient targeted therapies to be formulated. The AKT/mTOR pathway is a critical signaling axis for cell growth, survival, motility and metabolism in OC. Inhibitors of this pathway have shown positive outcomes in other cancers in the clinic. Therefore, this pathway might serve as an important therapeutic target and targeting this pathway singly or in combination with chemotherapeutic drugs or other targeted therapies might help in the prevention and treatment of OC. This review was an attempt to highlight the significance of the AKT/mTOR pathway in the development and progression of OC and also provide a summary of the natural and synthetic inhibitors of this axis, identified through various preclinical and clinical studies, for the prevention and treatment of OC. 

## 2. The AKT/mTOR Signaling Pathway

Many different cellular processes such as protein synthesis, autophagy, cell cycle regulation, glycogen metabolism, fatty acid synthesis, nutrient uptake, organization of nuclear proteins, regulation of different hallmarks of cancer like proliferation, survival, angiogenesis, invasion, migration and apoptosis of the cancer cells and the modulation of other signaling pathways such as the nuclear factor kappa-B (NF-κB), extracellular-signal-regulated kinase (ERK), Janus kinase (JAK)-signal transducer and activator of transcription (STAT), are all influenced by the AKT/mTOR signaling pathway [1,13,14,15,16,17] (Figure 1). The key proteins involved in the regulation of this pathway are phosphoinositide 3-kinase (PI3K), AKT and the mTOR proteins. When different growth factors and ligands such as integrins, receptor tyrosine kinases (RTKs), cytokine receptors and G protein-coupled receptors (GPCRs) bind to their respective receptors on the cell membrane, they generate a stimulus and cause the activation of the cell surface receptors, subsequently leading to the phosphorylation of PI3K [18,19,20,21,22,23].

Phosphoinositide 3-kinase (PI3K), a heterodimer, consists of mainly four classes, IA-PI3K, IB-PI3K, II-PI3K, and III-PI3K, among which class IA-PI3K plays the most crucial role in cancer progression and development [19,24,25]. PI3K helps in the phosphorylation of the D3 hydroxyl group present in the inositol ring of phosphatidyl inositols and catalyzes the transformation of membrane-bound phosphatidylinositol-(4,5)-bisphosphate (PIP2) to phosphatidylinositol-(3,4,5)-trisphosphate (PIP3) [26]. PIP3 acts as a secondary messenger that can be inactivated through the dephosphorylation by phosphatase and tensin homolog (PTEN). This protein mainly recruits two kinases that contain a pleckstrin homology domain (PH domain) to the membrane, i.e., phosphoinositide-dependent kinase 1 (PDK1) and serine/threonine kinase AKT [18,25]. 

AKT, a member of the AGC (cAMP-dependent protein kinase 1 (PKA)/ cGMP-dependent protein kinase (PKG)/ protein kinase C (PKC)) protein kinase family, consists of a PH domain at the N-terminal and a connecting hinge region at the C-terminal along with a kinase domain. The AKT protein resides in the cytoplasm of the cell in its inactive form. The PH domain of AKT has a high affinity towards PIP3. As already mentioned, PIP3 recruits AKT, and upon binding to AKT it causes the conformational change and exposure of the phosphorylation sites of AKT, which in turn activates the AKT protein. Studies have reported three isoforms of AKT, i.e., AKT1, AKT2 and AKT3. The AKT1 isoform mainly helps in cell survival and the inhibition of apoptosis [27,28,29,30,31,32,33]. The AKT2 isoform modulates the intrinsic mitochondrial pathway of apoptosis, metabolism, cell invasion and migration, while the third isoform, AKT3, functions in the regulation of the migration in tumor cells [13,34]. In a recent study conducted in our laboratory, we demonstrated the isoform-specific role of AKT in OC. The results suggested that the AKT1 and AKT2 isoforms were overexpressed in OC and silencing these isoforms reduced cell survival and caused cell cycle arrest at the G2/M phase besides inhibiting the expression of proteins involved in cell proliferation such as cyclooxygenase-2 (COX-2), cyclin D1 and cell survival such as survivin and the anti-apoptotic protein, B-cell lymphoma 2 (Bcl-2) [1].

In the case of cancers, the enhancement of AKT activity due to somatic mutations results in the impairment of the downstream elements of AKT. The activation of AKT causes the inhibition of proapoptotic proteins, Bcl-2-associated death promoter (Bad) and Bcl-2-associated X protein (Bax). The inhibition of Bad stops its activity to repress the action of anti-apoptotic B-cell lymphoma extra-large (Bcl-xL) protein, which in turn inhibits the process of apoptosis. AKT also inactivates caspases directly involved in apoptosis and forkhead box protein O1 (FOXO-1), which acts as a transcription factor and regulates the expression of proapoptotic genes, including Bcl-2-like protein 11 or Bim and Fas-ligand (FasL) [25,34,35]. AKT is also known to deregulate the activity of glycogen synthase kinase 3 β (GSK-3β) and FOXO, which ultimately leads to the upregulation of cyclin D1 [25,36,37,38,39]. Upon the activation of cyclin D1, it causes an augmentation in the expression of cyclin-dependent kinase (CDK) 4 and 6 to pass the G1-phase and enter the replication phase, i.e., the S-phase, thus enhancing proliferation. Moreover, AKT also regulates the p27 cytoplasmic localization, which plays a crucial role in tumor aggressiveness and metastasis [19,40].

One of the major downstream targets of AKT is the mTOR protein kinase. AKT phosphorylates tuberous sclerosis complex (TSC)-2 and inhibits its expression, which prevents it from activating Ras homolog enriched in brain (RHEB). As a result, the RHEB- Guanosine-5’-triphosphate (GTP) activate the mammalian target of rapamycin complex 1 (mTORC1) by binding to it [41]. Furthermore, mTORC1 triggers the activation of the eukaryotic translation initiation factor 4 (eIF4) complex, which ultimately leads to tumor progression, cell cycle progression and decreased apoptosis. The complete activation of AKT necessitates phosphorylation on serine residues in the C-terminal region by the mammalian target of rapamycin complex 2 (mTORC2). The sites of phosphorylation include serine (Ser)473 in AKT1, Ser474 in AKT2 and Ser472 in AKT 3. The reachability of the active sites governs the activity of mTORC1 and mTORC2, which are controlled by the mTOR associated proteins [42]. Such proteins, Dishevelled, Egl-10 and Pleckstrin (DEP) domain-containing mTOR-interacting protein (DEPTOR), and mammalian lethal with SEC13 protein 8 (mLST) interact with both the mTOR complexes, whereas the regulatory-associated protein of mTOR (RAPTOR), proline-rich AKT substrate of 40 kDa (PRAS40), mitogen-activated protein kinase-associated protein 1 (MAPKAP1) and the rapamycin-insensitive companion of mTOR (RICTOR) interact with mTORC1 and mTORC2, respectively. mTORC1 can regulate the different cell processes via the activation of S6. This protein also regulates translation through the 4E-binding protein 1 (4E-BP1). The PI3K protein and ribosomes with the PH-domain of its MAPKAP1 subunit play a vital role in the activation and the functioning of mTORC2 [43,44,45,46]. These events in the PI3K/AKT/mTOR signaling ultimately create an impact on the survival, protein synthesis, growth, lipid homeostasis, metabolism and cytoskeletal organization of cells.

The AKT/mTOR axis also regulates other essential processes in the cell. For instance, it regulates the deregulation of GSK-3β that is involved in the process of glycogen synthesis [47]. Besides, the components of this pathway also govern the expression of ATP citrate lyase (ACLY) that is known to be involved in fatty acid synthesis [15,48]. Furthermore, this pathway is also involved in the regulation of glucose-transport-involving proteins such as phosphatidylinositol-4-phosphate 5-kinase (PIP5K) and AS160, glycolysis-involving proteins such as hexokinase and 6-phosphofructo-2-kinase/fructose-2, 6-biphosphatase 2 (PFKFB2) and the organization of nuclear proteins via Lamin A [49,50,51,52]. Thus, the AKT/mTOR pathway is intrinsic to the regulation of several cellular processes and it plays an essential role in tumorigenesis.

## 3. The AKT/mTOR Pathway Activation in OC

The activation of the AKT/mTOR pathway or its components is implicated in the pathogenesis of different forms of OC, as shown in Table 1 [47,53,54,55,56,57,58,59,60,61,62,63,64,65,66,67,68,69,70,71,72,73,74,75,76,77,78,79,80,81,82,83,84,85,86,87,88,89,90,91,92,93,94]. The HNSCC tissues which could not be differentiated based on the source have also been included in the table. The investigations discussed in Table 1 provide a brief idea of the association of AKT/mTOR activation with oral tumorigenesis. For example, an immunohistochemical investigation revealed a greater expression of the activated forms of key proteins of the AKT/mTOR pathway such as AKT and mTOR in oral epithelial dysplasia (OED) than OSCC and non-dysplastic oral tissues (NDOT) [70]. In addition, the immunohistochemical analysis of mTOR in both human papillomavirus (HPV) (-) and HPV-associated HNSCC lesions have revealed that it is an important molecular target in OC [95]. The clinical significance of the AKT/mTOR axis has also been evidenced in human ameloblastoma tissues that showed a higher expression of p-AKT and p-mTOR compared to normal oral mucosa [96]. In 2018, Matsuo et al. reported the pathologic significance of AKT and mTOR in OSCC patients. He also hypothesized that the downstream protein of this axis, GSK3β, drove cervical lymph node (CLN) metastasis in OSCC patients [47]. Apart from the other malignancies associated with the oral cavity, a low-grade variant of OSCC, oral verrucous carcinoma (OVC) was also reported. The AKT/mTOR signaling was reported to be involved in the development and progression of OVC [88]. In quest of identifying prognostic and predictive markers for oral cancer, in 2017, Ferreira et al. studied the expression of several proteins in OSCC tissues. This study reported that the proteins, AKT and mTOR, were abundantly found in their active forms in OSCC tissues obtained from the gingiva, hard palate and alveolar ridge, suggesting that the activation of the AKT/mTOR pathway was associated with the development of OSCC [97]. These studies showed the significance of the AKT/mTOR pathway in the development and progression of OC. Thus, the overexpression or upregulation of this pathway induces tumor growth and results in poor prognosis [98]. 

## 4. Role of AKT/mTOR Pathway in Different Cellular Processes of OC Cells

The PI3K/AKT/mTOR pathway is altered in around 30.5% of HNSCC patients [99]. This pathway plays an important role in the proliferation, survival, invasion, angiogenesis, migration, protein synthesis and glucose metabolism of cells [100]. The gain of function mutations in the upstream gene, phosphatidylinositol-4,5-bisphosphate 3-kinase catalytic subunit alpha (PIK3CA), induces the oncogenic transformation of the AKT/mTOR signaling, leading to metabolic reprogramming in cells and subsequently causing OC development [101,102]. In 2016, Sonis and Mendes also postulated that the AKT/mTOR pathway drove the transformation of oral cells from benign to the malignant stage [103]. 

### 4.1. Proliferation and Survival

One of the most fundamental characteristics of malignant cells is the Warburg effect, which is characterized by enhanced aerobic glycolysis in cancer cells. Aberrant mTOR signaling pathway induces high glucose synthesis by lactic acid fermentation even when oxygen is sufficient (Warburg effect) which aids the survivial and progression of cancer cells [102]. For instance, the immunoregulatory protein B7-H3, also known as CD276, promoted aerobic glycolysis in OSCC cells via the AKT/mTOR aberration-induced augmented expression of hypoxia-inducible factor 1-alpha (HIF-1α), and subsequently, this led to the enhanced proliferation of OSCC cells [104]. The AKT/mTOR pathway also played a significant role in the stathmin-induced proliferation and survival of OSCC cells [105]. Similarly, a metformin derivative, HL156A, was found to inhibit OC cell viability and survival via the inhibition of insulin-like growth factor (IGF)-1/AKT/mTOR [106]. The anti-proliferative effect of Vietnamese coriander, also known as *Persicaria odorata*, and compound-like zerumbone was also reportedly mediated by the inhibition of the AKT/mTOR axis in OC cells [107,108]. Moreover, histone deacetylase inhibitors (HDACis) such as N-biphenyl-4-sulfonamide and 4-((biphenyl-4-ylsulfonyl) amino)-2-hydroxybenzamide, also showed antiproliferative activity in OC cells via interfering with the AKT/mTOR pathway [109]. Several studies also demonstrated that proteins such as the receptor for activated C kinase 1 (RACK1), dickkopf-related protein 3 (DKK3) and T-cadherin are crucial for the proliferation and survival of OSCC. These proteins regulate the proliferation and survival of OSCC via the activation of the AKT/mTOR pathway [71,110,111]. Besides, keratin 17 also displayed high expression in OC and this protein promoted tumorigenesis via the stimulation of AKT/mTOR signaling and glucose uptake [67]. 

### 4.2. Angiogenesis

The AKT/mTOR pathway plays an essential role in regulating the formation of blood vessels in both normal and cancer tissues. Angiogenesis is characterized by the secretion of vascular endothelial growth factor (VEGF), basic fibroblast growth factor (b-FGF) and interleukins such as interleukin-8 (IL-8) by cancer cells [112,113]. These cytokines bind to their receptors and activate the upstream molecules of the AKT/mTOR pathway. The PI3K activates the AKT/mTOR pathway that facilitates angiogenesis and the delivery of growth factors to tumors. On the contrary, cells with aberrant mTOR signaling also adapt and survive under nutrient-deprived and hypoxic conditions via another modified cellular response to hypoxia, nutrient uptake, etc. [114,115]. The role of AKT/mTOR signaling in the regulation of angiogenesis has also been observed in a hamster model of OC [116]. 

### 4.3. Epithelial-to-Mesenchymal Transition

The AKT/mTOR pathway also plays a remarkable role in the regulation of epithelial-to-mesenchymal transition (EMT) in cancer. Loss of PTEN is known to induce PI3K oncogenic disruption, which culminates in perturbation in membrane polarity, thereby leading to EMT that further induces invasion [117]. The involvement of AKT/mTOR signaling has been observed in the p70S6K-mediated promotion of IL-6-driven EMT and the metastasis of HNSCC [63]. Recently, Liu et al. also demonstrated that bone marrow mesenchymal stem cells aggravated tumorigenesis by activating an extracellular matrix (ECM) protein, periostin, that trigerred the activation of the PI3K/AKT/mTOR pathway, culminating in enhanced EMT in HNC cells [118]. In addition, the overexpression of epithelial cell adhesion molecules (EpCAMs) has also been reported to regulate EMT in HNC cells through the PTEN/AKT/mTOR pathway [119].

### 4.4. Invasion and Metastasis

Tumor outgrowths, induced by alterations in the AKT/mTOR pathway, lead to invasion [120]. Upon invasion, the PI3K–PTEN crosstalk controls the chemotaxis and intravasation of cells into endothelial networks [121]. This results in the AKT/mTOR-mediated regulation of angiogenesis, EMT, migration and invasion at the same time, subsequently causing metastasis. In a particular study, a neem limonoid, gedunin, in association with epalrestat, an aldose reductase inhibitor used for the treatment of diabetic neuropathy, was found to inhibit cell migration by inhibiting AKT/mTOR signaling in squamous cell carcinoma (SCC)-131 cells [122]. In another study, C-reactive protein (CRP) increased the invasion and migration of OSCC cells plausibly via the activation of the AKT/mTOR axis [123]. In 2019, Li et al. also reported the AKT/mTOR-mediated migratory and invasive potential of OSCC cells under the Warburg effect [104]. Moreover, studies have also shown the anti-invasive and anti-migratory potential of *P. odorat*a and zerumbone against OSCC cells, mediated by the modulation of AKT-mTOR signaling [107,108].

### 4.5. Autophagy

Autophagy is a process of lysosome-dependent cell degradation and the removal of damaged components of the cell along with intracellular pathogens [124,125]. It plays a significant role in cellular survival and is known to function via the modulation of the AKT/mTOR pathway in many cases. For instance, honokiol-treated OSCC cells underwent autophagy that was plausibly mediated via the downregulation of the AKT/mTORC1 signaling [126]. The involvement of AKT/mTOR signaling was also reported in Beclin-1-dependent autophagy in OSCC mediated via Tanshinone IIA, a lipophilic compound obtained from the roots of *Salvia miltiorrhiza* [127]. Similarly, another compound, resveratrol, was also found to exert autophagy in cisplatin-resistant CAR cells via the modulation of AKT/mTOR signaling [128]. Furthermore, the knockdown of neutrophil gelatinase-associated lipocalin (NGAL) activated mTOR and suppressed autophagy, thereby promoting the progression of OC. This study also suggested the involvement of the AKT/mTOR pathway in NGAL-mediated regulation of autophagy in OC cells [9]. 

### 4.6. Circadian Cock Signaling

The circadian clock signaling involves genes that maintain the circadian rhythm of the human body. These genes also interfere with the other cellular processes such as proliferation, apoptosis, cellular metabolism, cell cycle, immunity and endocrine signaling. Therefore, the deregulation of the clock signaling has been evidenced in various pathological conditions. The functioning of this signaling pathway requires the involvement of the AKT/mTOR pathway in OC [129,130]. For instance, the loss of circadian clock genes, Per1 and Per2, have been reported to increase the proliferation of OC cells and promote their progression by suppressing autophagy-induced apoptosis in an AKT/mTOR pathway-dependent manner [131,132]. These studies demonstrated the significance of the AKT/mTOR axis in circadian clock signaling.

### 4.7. Chemoresistance and Radioresistance

The increasing number of evidences suggest the pivotal role of the AKT/mTOR pathway in chemoresistance and radioresistance in cancer cells. Thus, the inhibition of this pathway might help in the reversal of chemoresistance and radioresistance, thereby making this pathway an attractive target for developing cancer therapeutics against OSCC. This pathway has been reported to be involved in chemosensitization mediated by a combination of chemotherapeutic drugs with other drugs. For example, prior treatment of chemoresistant oral epidermoid cancer cells with pantoprazole was found to chemosensitize these cells to vincristine both in vitro and in vivo via the inhibition of the AKT/mTOR pathway, among other related pathways [133]. Similarly, the anti-viral drug Ribavirin was reported to chemosensitize OSCC cells to paclitaxel via the inactivation of proteins such as AKT, mTOR, and eukaryotic translation initiation factor (eIF4E) 4E (4E-BP1) [134]. Additionally, Wang et al. also revealed that acetylshikonin considerably suppressed the growth of cisplatin-resistant OC both in in vitro cellular models and in vivo xenograft mice models by inhibiting the mTOR/PI3K/AKT signaling pathway [135]. In another preclinical study, the significant antitumor effect of a combination of mTOR inhibitor, temsirolimus and an anti-EGFR agent, cetuximab, was observed in an orthotopic model of HNSCC. The synergistic effect of this combination of drugs was also reportedly mediated via the inhibition of the PI3K/mTOR pathway [136]. 

Radioresistance is another phenomenon in cancer cells where the AKT/mTOR pathway plays a significant role. A study by Gu et al. indicated that tongue cancer resistance-associated protein 1 (TCRP1) mediates radioresistance in OSCC cells by elevating AKT activity and NF-κB level [137]. In 2014, Freudlsperger et al. demonstrated that the inhibition of AKT (Ser473) phosphorylation might overcome radioresistance, thereby decreasing toxicity and ameliorating the efficiency of treatment in advanced HNSCC [138]. Another study by Yu et al. evaluated the efficacy of a second generation mTOR inhibitor, AZD2014, also known as Vistusertib, as a radiosensitizing agent in primary OSCC and OSCC-derived cell lines. The co-treatment of irradiated OSCC cells with AZD2014 exhibited a synergistic dual blockade of mTORC1 and mTORC2/AKT activity and cell cycle arrest, leading to cell-growth inhibition and radiosensitization of the OSCC cells [139]. In 2017, Yu et al. demonstrated that the activation of PI3K/AKT/mTOR signaling contributed to radioresistance in OSCC. This study reported that the dual inhibition of the PI3K/mTOR axis led to the inhibition of cyclin D1/CDK4 activity, thereby inducing G1 phase arrest in OC samples [140]. Thus, the AKT/mTOR pathway is intrinsic to the regulation of chemoresistance and radioresistance in OC cells.

## 5. MicroRNA (miRNA), Long Noncoding RNA (lncRNA), and Circular RNA (circRNA) Regulate AKT/mTOR Pathway in OC

### 5.1. MicroRNAs

MicroRNAs (miRNAs) are a class of endogenous, short noncoding RNAs that are highly conserved. They regulate various critical biological processes and are known to be dysregulated in several forms of cancer [141,142]. miRNAs play a prominent role in tumor progression by directly targeting multiple genes. Thus, the investigation of aberrantly expressed miRNAs may play a crucial role in the diagnosis and treatment of OSCC or HNSCC [141,142,143]. The miRNA profiling of primary OSCC tissue specimens has revealed that 46 differentially expressed miRNAs have activated PI3K/AKT signaling genes and disrupted p53 signaling pathways [144]. Genome-wide profiling and functional assays of miRNA-99 (miR-99)-transfected HNSCC cells have revealed that mTOR is the direct target gene of miR-99 and that the increased expression of miR-99 in HNSCC downregulated the expression of mTOR genes [145]. miR-99a was also found to be associated with the enhanced tumor size in OSCC. This miRNA was also reported to be involved in AKT/mTOR signaling [146]. Moreover, miR-218 suppressed the expression of RICTOR, which further inhibited AKT S473 phosphorylation in OSCC [147]. The overexpression of miR-27a* (miR-27a-5p) in HNSCC cells has exhibited a significant cytotoxic effect compared to miR-27a-3p, and it has been found to downregulate AKT1 and mTOR by direct inhibition [148]. In another study, miR-491-3p sensitized OC cells to chemotherapy through the inhibition of mTORC2 activity by directly targeting mTORC2 component RICTOR, and it was upregulated in the drug-resistant OC cells with elevated levels of p-AKT(Ser473), p-FOXO-1(Thr24) and phospho-serum/glucocorticoid regulated kinase 1 (p-SGK1) (Ser422) [149]. Thus, miRNAs play an important role in the regulation of the AKT/mTOR pathway.

### 5.2. Circular RNAs

Circular RNAs (circRNAs) are another type of non-coding single-stranded closed RNA molecules that are generated as covalently closed continuous loops. Studies have reported that the activation of circRNAs induces tumor migration and invasion in several cancers. Under hypoxic conditions, circCDR1as, a circRNA antisense to the cerebellar degeneration-related protein 1 (CDR1) transcript, was found to enhance autophagy and promote the survival of OC cells by inhibiting apoptosis through the regulation of AKT/ERK1/ERK2/mTOR signaling [150]. The AKT/mTOR signaling pathway played a significant role in the hsa_circ_0007059-mediated regulation of OC cell growth. The elevated expression of hsa_circ_0007059 was found to suppress proliferation, migration and invasion, as well as induce apoptosis in OSCC cells [151].

### 5.3. Long Noncoding RNAs 

Long noncoding RNAs (lncRNA) belong to another class of noncoding RNAs that have a length of more than 200 nucleotides [152,153,154,155]. Congregate evidence has shown that the different anomalous expression of lncRNA is closely associated with the manifestation of several diseases, including tumors [156,157,158]. In 2019, Yang et al. showed the oncogenic effect of lncRNAs in OC. This study demonstrated that the knockdown of lncRNA cancer susceptibility candidate 9 (CASC9) suppressed tumor progression by inhibiting the proliferation of OSCC cells and promoting autophagy-mediated cell apoptosis via the modulation of AKT/mTOR pathway [159]. Similarly, the silencing of the lncRNA homeobox(HOX) transcript antisense RNA (HOTAIR) induced the suppression of the autophagy in OC cells by promoting the activation of mTOR. The knockdown of this lncRNA was reported to increase the rate of apoptosis and enhance the sensitivity of OC cells to cisplatin [160]. 

## 6. Therapeutic Effect of AKT/mTOR Inhibitors in OC

The corroboration of the preclinical and clinical studies have implicated that AKT/mTOR inhibitors are emerging therapeutics for the treatment of various cancers. The encouraging outcomes of in vitro and in vivo investigations have instigated the initiation of several clinical trials of the AKT/mTOR inhibitors to determine their safety and efficacy against HNC. These inhibitors have shown remarkable prospects in the treatment of OC or HNC. The efficacy of these inhibitors in both preclinical and clinical settings is presently discussed.

### 6.1. Inhibitors in Preclinical Studies

It is increasingly evident from several in vitro and in vivo studies that the AKT/mTOR pathway is remarkably involved in the management of OC. Several inhibitors of this pathway, both natural and synthetic, have been identified in preclinical settings. In silico studies have also envisioned the clinical significance of natural inhibitors of AKT [161]. Huge evidence emerging from several investigations has consistently emphasized the AKT/mTOR-dependent anticancer activity of the natural compounds such as piceatannol, boswellic acid, curcumin, honokiol, magnolol, tocotrienol, capsaicin, diosgenin, garcinol, thymoquinone and gambogic acid [162,163,164,165,166,167,168,169,170,171]. Additionally, extracts from plants like *Punica granatum* and *P. odorata* have also shown to abrogate this pathway [107,172]. The inhibitors of the AKT/mTOR pathway, identified from preclinical studies, such as compounds like acetylshikonin, artesunate, arglabin, AZD2014, ellagic acid, erfosin, fenofibrate, HL156A, mecambridine, murrayanine, oleic acid, pantoprazole, vincristine, S-allylcysteine, tanshinone IIA, and ursolic acid, PI3K inhibitors such as PI-828, PI-103 and PX-866 and miRNAs such as miR-218 have been listed in Table 2 [106,107,127,133,135,139,147,173,174,175,176,177,178,179,180,181,182,183]. The artesunate-mediated suppression of mitochondrial respiration via AKT/AMP-activated protein kinase (AMPK)/mTOR inhibition was reported in in vitro and in vivo OSCC models [174]. In addition, the targeted inhibition of the AKT/mTOR axis was also obtained with PI3K inhibitors such as PI-828, PI-103 and PX-866 (Sonolisib) in OC cells SCC-4, SCC-9 and SCC-25 [181]. Furtheremore, several other compounds like honokiol, plumbagin, and small molecules such as pyrithione zinc (PYZ) have also demonstrated anti-cancer effects via the inhibition of the AKT/mTOR pathway in OC [126,184,185]. Moreover, the golden nutraceutical, curcumin, which is known to show potent anticancer effects, was found to inhibit the nicotine-induced activation of the AKT/mTOR pathway in OC models in vitro and in vivo [12]. Furthermore, other compounds like escin, zerumbone, oxymatrine and formononetin also demonstrated anti-tumor effects via the inhibition of the AKT/mTOR pathway in several preclinical models [186,187,188,189,190]. Thus, these studies led to the evaluation of AKT/mTOR inhibitors in clinical settings.

### 6.2. Inhibitors in Clinical Studies

The number of clinical trials determined to evaluate the efficacy of the inhibitors of AKT, mTOR, or the AKT/mTOR pathway in OC patients is very few. However, several clinical trials have been reported with details of such investigation in HNSCC patients (Table 3) [191,192,193,194,195,196,197,198,199,200,201,202,203,204,205]. Some of these trials have been completed or terminated, while some are still ongoing. The drugs used in clinical trials, for HNC, either as monotherapy or combinatorial therapy, are discussed below. 

#### 6.2.1. Bimiralisib

Bimiralisib is mainly a PI3K inhibitor and to some extent, it also inhibits the mTOR kinase. The safety and efficacy of this antineoplastic compound is being evaluated in an ongoing clinical trial in patients with recurrent or metastatic (R/M) HNSCC that bear Notch homolog 1, translocation-associated (NOTCH1) loss of function (LOF) mutations (NCT03740100). The NOTCH1 protein plays a central role in the maintenance of stem cells and determination of the fate of the cells. Studies in the recent past have implicated LOF in NOTCH1 genes in HNSCC samples [206,207].

#### 6.2.2. CC-115

CC-115 is a selective dual mTOR/DNA-dependent protein kinase (DNA-PK) inhibitor. Currently, many clinical trials are undergoing to test the safety and efficacy of this compound in various cancers. In a recent investigation involving HNSCC tumors, this compound has exhibited good tolerability and promising effects (NCT01353625) [208].

#### 6.2.3. Everolimus

Everolimus, also known as Rad001, is an mTOR kinase inhibitor and it has been widely used in HNC patients (NCT01051791, NCT01111058; NCT01133678) [192]. Everolimus was not found to be effective in the form of monotherapy and adverse effects were observed (NCT01051791) [192]. Therefore, several clinical trials were conducted in HNC patients where everolimus was administered in combination with other drugs such as cetuximab, erlotinib, carboplatin and cetuximab, carboplatin and paclitaxel, cisplatin and docetaxel, cisplatin or carboplatin and cetuximab and cisplatin and radiotherapy. Most of these studies have been completed [193,194,195,196] (NCT01637194, NCT00942734, NCT01283334, NCT01333085, NCT00935961) while a few combinations have been discontinued due to severe side effects (NCT01009346, NCT01057277). However, the regime (everolimus, cisplatin and radiotherapy) was well tolerated in HNC patients with promising efficacy (NCT00858663) [197]. 

#### 6.2.4. Metformin

The antidiabetic drug, metformin, which also possesses the ability to suppress tumor growth, is known to inhibit the PI3K/AKT signaling pathway [209]. This drug is also known to minimize oxygen consumption in cells via the suppression of mitochondrial complex I [210]. Owing to this property, the PI3K/AKT inhibitor, metformin, is being administered to HNSCC patients suffering from tissue hypoxia in order to evaluate the effect of metformin upon hypoxic conditions in tumor (NCT03510390).

#### 6.2.5. MK2206

MK2206 is one of the few AKT inhibitors that have been used against HNC to date. This molecule is an allosteric inhibitor that inhibits the phosphorylation of AKT at sites threonine (T308) and serine (S473). This molecule has been evaluated as a drug for the treatment of R/M adenoid cyst carcinoma of oral cavity and salivary gland and recurrent/advanced SCC of the nasopharynx (NCT01604772, NCT01349933) [198]. However, these studies did not show positive outcome, and serious adverse effects such as constipation, dysphagia, gastritis, stomach ache, vertigo, vomiting, fatigue, fever, chest pain, dermal infections, increased levels of liver enzymes, hyperglycemia, flank pain, epistaxis, hiccups, dizziness and maculopapular rashes were noted.

#### 6.2.6. Perifosine

Perifosine, an oral alkylphospholipid, is another AKT inhibitor that inhibits the phosphorylation of AKT at sites threonine (T308) and serine (S473). This molecule has been used against R/M HNSCC as monotherapy, but severe adverse effects such as abnormal bilirubin, infection, fatigue, abnormal platelet count, hyponatremia, hypercalcemia and anorexia were reported upon the use of perifosine [199].

#### 6.2.7. Rapamycin

Rapamycin is a drug that selectively inhibits mTORC1 and impairs cancer metabolism. Phase 1 and 2 trials were carried out in stage II-IVA HNSCC patients to test the safety and efficacy of rapamycin. This compound was well tolerated and showed promising effects in HNSCC patients (NCT01195922) [200]. However, in combination with bevacizumab, rapamycin did not show any objective response (OR) in HNSCC patients [211].

#### 6.2.8. Ridaforolimus

Ridaforolimus is a small molecule mTOR inhibitor and an analog of rapamycin with significant anticancer activity. A phase I study was conducted to evaluate the combined activity and safety profile of mTOR inhibitor, ridaforolimus, and γ-secretase inhibitor, MK-0752. This study showed that the combination of ridaforolimus and MK-0752 showed a potential antitumor effect against HNSCC. However, many adverse events were reported at the maximum tolerated dose, which would require cautious management in the course of future clinical development (NCT01295632) [201]

#### 6.2.9. SF1126

SF1126 is an inhibitor of PI3 kinase and mTOR. It is a conjugate of LY294002 connected to an Arg-Gly-Asp-Ser (RGDS) tetrapeptide. The drug permeability of this prodrug is enhanced as it can bind to specific integrins inside the tumor. A phase 2 trial was conducted in metastatic squamous neck cancer patients with occult primary squamous cell carcinoma (SCC) to determine the efficacy of SF1126 as monotherapy, but this study was terminated due to slow enrollment (NCT02644122).

#### 6.2.10. Temsirolimus

Temsirolimus is a water-soluble analog of rapamycin that explicitly inhibits mTOR. In patients with advanced stage HNSCC, temsirolimus suppresssed the mTOR pathway in tumors and peripheral blood mononuclear cell (PBMCs) of HNSCC with minimal side effects [212]. In 2015, Grunwald et al. assessed the efficacy of this drug in R/M HNSCC refractory to platinum and cetuximab. Of the patients involved in the study, 39.4% exhibited tumor shrinkage within the first six weeks of administration, though without any OR (NCT01172769) [202]. On the contrary, in another study, temsirolimus was used in combination with carboplatin and paclitaxel in R/M HNSCC patients and it resulted in an objective PR from 22% of the patients (NCT01016769) [203]. 

Temsirolimus along with erlotinib were also administered to R/M HNSCC patients, but this duo was reportedly toxic and fatal. Hence, the trial was terminated at an early stage (NCT01009203) [204]. In another trial, a combination of temsirolimus, bevacizumab and cetuximab was examined in advanced malignancies. The results showed that 25% of the patients with HNSCC achieved a PR while few patients withdrew as they started showing severe toxicities [213].

As mentioned previously, several inhibitors of the AKT/mTOR pathway have been evaluated as drugs in clinical settings. Despite a significant response in some cases, the use of these commercial synthetic inhibitors face major limitations in clinical settings [214,215,216]. For instance, the use of everolimus in cancer showed several side effects such as elevated alanine aminotransferase levels, stomatitis, hyperglycaemia, anaemia and pneumonitis [217]. In another study, temsirolimus has shown potential toxicity; however, side effects such as pneumonia, anemia and fatigue were also reported in the HNSCC patients [25]. As already mentioned, the AKT inhibitors perifosine and MK2206 also caused remarkably adverse effects. Other severe adversities caused by rapamycin, temsirolimus and everolimus included blood count abnormalities, head and neck edema, gastrointestinal disorders, increased liver enzymes, hypokalemia, hyperglycemia, hyponatremia, pruritus, mucositis, infections, nervous system disorders and fatigue [25]. Many clinical studies have been terminated due to such toxicities. To facilitate the broad-spectrum utilization of targeted inhibitors in the clinical scenario, the toxicity of these inhibitors should be minimized. Furthermore, natural inhibitors provide a safe, efficacious and an inexpensive alternative to the commercial inhibitors. 

## 7. Conclusions

OSCC, the primary subtype of HNSCC, is a significant health issue that comprises of cancers of the oral cavity. Despite aggressive treatment methods, the five-year survival rate of OC is low. Several studies have identified major alterations in the genetic makeup of OSCC samples. However, the actual mechanism of oral tumorigenesis has not yet been deciphered. In order to alleviate the survival rate of OC patients, targeted therapy is pursued as an emerging regimen. Among several proteins and their associated signaling pathways, the AKT/mTOR pathway is one of the most altered axes reported in OC to date. Therefore, inhibitors of the AKT/mTOR pathway are sought for the prevention and treatment of OC. The preclinical studies have shown significant alteration of this axis in OC and targeting this axis by several natural and synthetic inhibitors has shown the modulation of various cellular processes such as proliferation, survival, angiogenesis, invasion, metastasis, autophagy, epithelial to mesenchymal transition, circadian clock signaling, chemoresistance and radioresistance. miRNAs, circRNAs and lncRNAs have also demonstrated an active role in the regulation of the AKT/mTOR axis in OSCC patients. Preclinical and clinical studies have illustrated the efficacy of certain inhibitors and their combination with other standard chemotherapeutic drugs such as paclitaxel, cisplatin or other inhibitors such as EGFR inhibitor or cetuximab in the prevention and treatment of OC. However, the use of these inhibitors has shown adverse effects in some cases, which need to be overcome for the persistent use of clinical inhibitors for the treatment of OC. Thus, AKT/mTOR inhibitors seem to hold great potential in the management of OC and tailoring these inhibitors for suitable use in the clinic might help to design personalized therapy for OC patients.

## Figures and Tables

**Figure 1 ijms-21-03285-f001:**
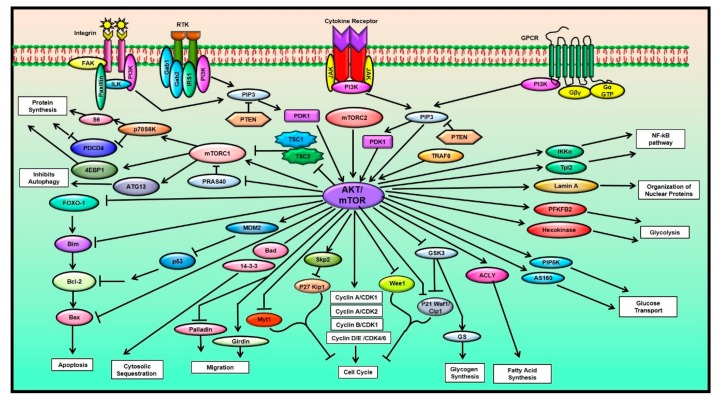
The AKT/mTOR signaling and its role in various cellular processes (Abbreviations: 4E-BP1: Eukaryotic translation initiation factor 4E (eIF4E)-binding protein 1; ACLY: ATP citrate lyase; AS160: Akt substrate of 160 kDa; ATG13: Autophagy-related protein 13; Bad: Bcl2-associated agonist of cell death; Bax: Bcl 2-associated X protein; Bcl-2: B-cell lymphoma 2; CDK: Cyclin-dependent kinase; Cip1: CDK-interacting protein 1; FAK: Focal adhesion kinase; FOXO-1: Forkhead box protein O1; Gab1: GRB2-associated-binding protein 1; Gab2: GRB2-associated-binding protein 2; Girdin: Girders of actin filament; GPCR: G protein-coupled receptor; GS: Glycogen synthase; GSK3: Glycogen synthase kinase-3; Gα GTP: GTP-bound Gα subunit; Gβγ: G beta-gamma complex; IKKα: IκB Kinase α; ILK: Integrin-linked kinase; IRS1: Insulin receptor substrate 1; JAK: Janus kinase; MDM2: Mouse double minute 2 homolog; mTOR: Mammalian target of rapamycin; mTORC1: Mammalian target of rapamycin complex 1; mTORC2: Mammalian target of rapamycin complex 2; Myt1: Myelin transcription factor 1; P70S6K: 70 kDa ribosomal protein S6 kinase; PDCD4: Programmed cell death protein 4; PDK1: Phosphoinositide-dependent kinase 1; PFKFB2: 6-Phosphofructo-2-Kinase/Fructose-2,6-Biphosphatase 2; PI3K: Phosphoinositide 3-kinase; PIP3: Phosphatidylinositol-(3,4,5)-trisphosphate; PIP5K: Phosphatidylinositol-4-phosphate 5-kinases; PRA S40: Proline-rich Akt substrate of 40 kDa; PTEN: Phosphatase and tensin homolog; RTK: Receptor tyrosine kinases; S6: Ribosomal protein S6; Skp2: S-Phase kinase-associated protein 2; Tpl2: Tumor progression locus 2; TRAF6: Tumor necrosis factor receptor-associated factor 6; TSC1: Tuberous sclerosis protein 1; TSC2: Tuberous sclerosis protein 2; Wee1: Wee1-like protein kinase).

**Table 1 ijms-21-03285-t001:** List of the studies showing the activation of AKT, mTOR or the AKT/mTOR pathway in OC.

Type of Cancer	In vitro/ In vivo/ Ex vivo	Model	References
HNSCC	In vivo	Grhl3^+/–^ and Grhl3^∆/–^ /K14Cre^+^ (cKO) mice	[53]
	In vitro	SCC-25, CAL-27 cells	[53]
	In vitro	HSC-3 cells	[54]
	In vitro In vitro In vitro	CAL-27, HN30 cells CAL-27 cells HN6 cells	[55] [56] [57]
	In vivo	Tgfbr1 cKO mice	[58]
	In vitro	NOK-SI cells having SET protein overexpression	[59]
	Ex vivo	Primary tumor tissue from patients	[60]
	In vitro	HN5 cells	[61]
	In vitro	FaDu, SAS cells	[62]
	In vitro	IL-6 treated 686LN cells	[63]
	In vitro	HGF stimulated and treated UT-SCC-14, UT-SCC-15 and	[64]
		UT-SCC-16A cells-injected mice xenografts	
	In vitro In vitro In vitro	PCI-9A, PCI-15 cells WSU-HN6, CAL27 cells Ca9-22, HSC-3 cells	[65] [66] [67]
OED	In vitro	DOK cells	[68]
OED	In vitro	DOK cells	[69]
OED	Ex vivo	Tissue from patients	[70]
OED	Ex vivo	Tissue from patients	[71]
OED	Ex vivo	Tissue from patients	[72]
OED	Ex vivo	Tissue from patients	[73]
OEPL	Ex vivo	Tissue from patients	[74]
OL	Ex vivo	Tissue from patients	[72]
OPSCC	Ex vivo	Primary tumor tissue from patients	[75]
OSCC	Ex vivo	Tissue from patients	[70]
OSCC	Ex vivo	Tissue from patients	[47]
OSCC	In vivo	Keratin 17-knockout HSC3 cells injected BALB/c mice	[67]
OSCC	In vitro	HSC-3, HSC-4, CAL-27, UM1, UM2 cells	[71]
OSCC	Ex vivo	Tissue from patients	[71]
OSCC	Ex vivo	Tissue from patients	[76]
OSCC	In vitro	Tca-8113, KB cells	[77]
OSCC	In vitro	SCC-25, SCC-4 cells	[78]
OSCC	In vitro	HSC-6, CAL-33 cells	[79]
OSCC	In vitro	SAS, OECM-1 cells	[80]
OSCC	In vitro	OECM-1 cells	[72]
OSCC	Ex vivo	Tissue from patients	[72]
OSCC	In vitro	KB cells	[81]
OSCC OSCC OSCC	In vitro In vitro In vitro	SAS cells SCC-9, SCC-25 cells UM-SCC-22A cells	[82] [68] [69]
OSCC	In vitro	OECM-1 cells	[83]
OSCC	Ex vivo	Buccal mucosa and other tissues (Stage:1-4, Grade:1, 2 or 3)	[83]
OSCC	In vitro	SCC-4, CAL-27 cells	[84]
OSCC	Ex vivo	Tissue from patients	[84]
OSCC	Ex vivo	Tissue from patients	[85]
OSCC	In vitro	AW13516 cells	[86]
OSCC	Ex vivo	Primary tumor tissues from patients (Stage: 1-4)	[87]
OSCC	Ex vivo	Tissue from patients	[88]
OSCC	Ex vivo	Tissue from patients	[74]
OVC	Ex vivo	Tissue from patients	[72]
OVC	Ex vivo	Tissue from patients	[88]
TC TD TSCC TSCC TSCC	Ex vivo Ex vivo Ex vivo In vitro Ex vivo	Tissue from patients (Early stage) Tissue from patients Tissue from patients SCC-4, SCC-25 Tissue from patients	[73] [89] [89] [90] [91]
TSCC	In vitro	CAL-27 cells	[92]
TSCC	In vitro	UM1 cells	[93]
TSCC	In vitro	CAL-27 cells and cisplatin resistant Tca cells	[94]

**Abbreviations:** cKO: Conditional knock out, Cre: Cre recombinase, DOK: Dysplastic oral keratinocyte, Grhl3: Grainyhead-like 3, HGF: Hepatocyte growth factor, HNSCC: Head and neck squamous cell carcinoma, K14: Keratin 14, NOKI-SI: Normal Oral Keratinocytes spontaneously immortalized, NOM: Normal oral mucosa, OC: Oral Cancer, OED: Oral epithelial dysplasia, OEPL: Oral epithelial precursor lesions, OL: Oral leukoplakia, OPSCC: Oropharyngeal squamous cell carcinoma, OSCC: Oral squamous cell carcinoma, OVC: Oral verrucous carcinoma, SCC: squamous cell carcinoma, SCID: Severe Combined Immunodeficiency, TC: Tongue cancer, TD: Tongue dysplasia, Tgfbr1: TGF-β receptor I, TSCC: Tongue squamous cell carcinoma.

**Table 2 ijms-21-03285-t002:** List of preclinical studies showing mechanism of action of several AKT/mTOR inhibitors in OC

Inhibitor	Model	Mechanism of Action	Reference
Acetylshikonin	Both	↓mTOR/PI3K/AKT pathway, p62, Bcl-2; ↑Beclin-1, LC3-II, Bax	[135]
Arglabin	Both	↓mTOR/PI3K/AKT, Δψm; ↑ROS	[173]
Artesunate	Both	↓AKT/AMPK/mTOR; ↑Mitochondrial dysfunction, ROS	[174]
AZD2014	In vitro	↓AKT, mTORC1, mTORC2, cyclin D1-CDK4, cyclin B1-CDC2; ↑caspase-3, LC3	[139]
Ellagic acid	In vivo	↓PI3K/AKT/mTOR, MAPK, VEGF/VEGFR2, HDAC6, HIF-1α	[175]
Erufosine	In vitro	↓AKT/mTOR, Cyclin D1	[176]
Fenofibrate	In vitro	↓AKT, mTOR, RAPTOR; ↑AMPK	[177]
HL156A	Both	↓AKT, mTOR, IGF-1, ERK1/2, NF-κB-p65, MMP-2, MMP-9↑AMPK, ROS, caspase-3 and -9, p-AMPK	[106]
Mecambridine	In vitro	↓mTOR/PI3K/AKT signaling, MMP; ↑ROS	[178]
miR-218	In vitro	↓mTOR-AKT signaling pathway, RICTOR	[147]
Murrayanine	In vivo	↓AKT/mTOR and Raf/MEK/ERK pathways; ↑caspase-3, Bax/Bcl-2	[179]
Oleic acid	In vitro	↓p-AKT, p-mTOR, p-S6K, p-4E-BP1, p-ERK1/2, Cyclin D1, LC3-I/ LC3-II, p62	[180]
*Persicaria odorata*	In vitro	↓AKT/mTOR pathway, Cyclin D1, COX-2, survivin, MMP-9, VEGF-A	[107]
PI-103, PI-828, PX-866	In vitro	↓AKT, mTOR, COX-2, Cyclin-D1, VEGF, PI3Kα, Bcl-2, NF-κB	[181]
PPZ + VCR	Both	↓PI3K/AKT/mTOR pathway	[133]
S-Allylcysteine	In vivo	↓pAKT, mTOR, IκBα, ERK1/2, Cyclin D1, ↓NF-κB p65,↓COX-2, ↓vimentin; ↑E-cadherin, p16	[182]
Tanshinone IIA	In vitro	↓PI3K/AKT/mTOR pathway; ↑Beclin-1/ATG7/ATG12-ATG5	[127]
Ursolic acid	In vitro	↓AKT/mTOR/NF-κB signaling, ERK, p38, MMP-2; ↑Caspases	[183]

**Abbreviations:** ↑: Upregulation, ↓: Downregulation, Δψm: Mitochondrial membrane potential, 4E-BP1: Eukaryotic translation initiation factor 4E (eIF4E)-binding protein 1, AMPK: AMP-activated protein kinase, ATG: Autophagy related, Bad: Bcl-2-associated agonist of cell death, Bax: Bcl-2-associated X protein, Bcl-2: B-cell lymphoma 2, Bcl-xL: B-cell lymphoma-extra-large, CDC2: Cell division cycle protein 2 homolog, CDK4: Cyclin-dependent kinase 4, COX-2: Cyclooxygenase 2, ERK: Extracellular-signal-regulated kinase, HDAC6: Histone deacetylase 6, HIF-1α: hypoxia-inducible factor 1-alpha, IGF-1: Insulin-like growth factor 1, IκBα: Inhibitor of nuclear factor kappa B, MAPK: Mitogen-activated protein kinase, LC3: Microtubule-associated protein 1A/1B-light chain 3, MEK: Mitogen-activated protein kinase, MMP: Matrix metalloproteinase, mTOR: Mammalian target of rapamycin, NF-κB: nuclear factor kappa-B, PI3K: Phosphoinositide 3-kinase, PPZ: Pantoprazole, pS6K: phosphorylated S6K, RAPTOR: Regulatory-associated protein of mTOR, RICTOR: Rapamycin-insensitive companion of mTOR, ROS: Reactive oxygen species, VCR: Vincristine, VEGF: Vascular endothelial growth factor.

**Table 3 ijms-21-03285-t003:** List of clinical trials showing use of inhibitors of AKT, mTOR or AKT/mTOR in HNC patients.

Inhibitor	Combinatorial Therapy	Phase	Status	Target	Sample	NCT/REF	Reference
Bimiralisib	-	2	Recruiting	PI3K/mTOR	HNSCC harboring NOTCH1 LOF mutation	NCT03740100	-
CC-115	-	1	Active, not recruiting	AKT/mTOR	HNSCC	NCT01353625	
Everolimus	-	2	Completed	mTOR	R/M HNSCC	NCT01051791	[192]
Everolimus	-	2	Active, not recruiting	mTOR	HNC	NCT01111058	
Everolimus	-	1+2	Active, not recruiting	mTOR	LA-HNSCC	NCT01133678	
Everolimus	Cetuximab	1	Completed	mTOR	Recurrent HNC	NCT01637194	
Everolimus	Erlotinib	2	Completed	mTOR	Recurrent HNSCC	NCT00942734	[193]
Everolimus	Carboplatin + cetuximab	1+2	Completed	mTOR	Advanced HNSCC	NCT01283334	[194]
Everolimus	Carboplatin + paclitaxel	1+2	Completed	mTOR	LA-HNSCC	NCT01333085	[195]
Everolimus	Cisplatin + docetaxel	1	Completed	mTOR	LA-HNSCC	NCT00935961	[196]
Everolimus	Cisplatin/carboplatin+ cetuximab	1+2	Terminated	mTOR	R/M HNSCC	NCT01009346	
Everolimus	Cisplatin + radiotherapy	1	Terminated	mTOR	LA inoperable HNC	NCT01057277	
Everolimus	Cisplatin + radiotherapy	1	Completed	mTOR	HNC	NCT00858663	[197]
Metformin	-	-	Recruiting	PI3K/AKT	OSCC	NCT03510390	
MK2206	-	2	Completed	AKT	Progressive, R/M of oral cavity and SG	NCT01604772	[198]
MK2206	-	2	Completed	AKT	Recurrent/advanced SCC of nasopharynx	NCT01349933	
Perifosine	-	2	Completed	AKT, PI3K	R/M HNSCC		[199]
Rapamycin	-	1+2	Completed	mTOR	Stage II-IVA HNSCC	NCT01195922	[200]
Ridaforolimus	MK-0752	1	Completed	mTOR	Metastatic or LA- HNSCC	NCT01295632	[201]
SF1126	-	2	Terminated	PI3K, mTOR	Metastatic HNSCC	NCT02644122	
Temsirolimus	-	2	Completed	mTOR	Platinum/cetuximab- refractory HNSCC	NCT01172769	[202]
Temsirolimus	Cetuximab	2	Completed	mTOR	R/M HNSCC	NCT01256385	
Temsirolimus	Carboplatin + paclitaxel	1+2	Completed	mTOR	R/M HNSCC	NCT01016769	[203]
Temsirolimus	Erlotinib	2	Terminated	mTOR	R/M P-R HNSCC	NCT01009203	[204]
Temsirolimus	Paclitaxel+ low dose weekly carboplatin	1+2	Completed	mTOR	R/M HNSCC		[205]

**Abbreviations:** ACC: Adenoid cyst carcinoma, HNC: Head and neck cancer, HNSCC: Head and neck squamous cell carcinoma, LA: Locally advanced, LA-SCCHN: Locall advanced squamous cell carcinoma of the head and neck, mTOR: Mammalian target of rapamycin, OSCC: Oral squamous cell carcinoma, PI3K: Phosphoinositide 3-kinase, P-R: Platinum refractory, R/M: Recurrent or metastatic, SCC: Squamous cell carcinoma, SG: Salivary gland.

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
