# Peer review of "Targeting AKT/mTOR in Oral Cancer: Mechanisms and Advances in Clinical Trials"

_ijms, 2020, doi:10.3390/ijms21093285_

Round 1
Reviewer 1 Report
In the present manuscript, Authors have discussed about the role of AKT/mTOR pathway in the development of oral cancer. They have mentioned the basic biology of AKT/mTOR pathway. The review includes preclinical data and clinical trials data identifying the importance of AKT/mTOR pathway in oral cancer. Authors have summarized the main achievements in the reviewed field, and pointed out the gaps in oral cancer therapy, such as using EGFR therapy and synthetic inhibitors. Overall, the review supports the use of mTOR inhibitors for the treatment of oral cancer. However, there are minor issues associated with publication:
- Under section “Role of AKT/mTOR pathway in proliferation, survival, invasion, angiogenesis, autophagy, metastasis and EMT in OC”, authors should consider addressing the role of clock genes as important targets in oral cancer. Liu et al. (2020) and Yang et al. (2020) have reported the involvement of Per1 and Per2 genes in oral squamous cell cancer suppression by promoting autophagy via AKT/mTOR pathway.
- Line 171-172- Please check the sentence “In pursuit of identifying prognostic and predictive markers for this disease”.
Author Response
Dear Sir/Ma'am,
We are grateful to you for your insightful comments. These comments are very useful for strengthening the readability and increasing the significance of the manuscript. We have now modified the manuscript as per the comments. A point by point response to the comments is mentioned below.
Comment 1:
Under section “Role of AKT/mTOR pathway in proliferation, survival, invasion, angiogenesis, autophagy, metastasis and EMT in OC”, authors should consider addressing the role of clock genes as important targets in oral cancer. Liu et al. (2020) and Yang et al. (2020) have reported the involvement of Per1 and Per2 genes in oral squamous cell cancer suppression by promoting autophagy via AKT/mTOR pathway. (259-262) add references
Response: These references have been included in a new section ‘4.5. Circadian cock signaling’ along with a brief description of the clock genes. Please refer to the highlighted portion from line number 287-295, page no. 8.
Comment 2:
Line 171-172- Please check the sentence “In pursuit of identifying prognostic and predictive markers for this disease”.
Response: This sentence has been corrected. Please refer to the highlighted portion from line number 194-199, page no. 5.
Also please find the attachment
Reviewer 2 Report
AKT/mTOR signaling underlies many fundamental cell processes, and its mis-regulation is implicated in a variety of diseases. This review focuses to discuss the significance of AKT/mTOR signaling in oral cancer and its potential as a therapeutic target. This topic is of great interest and could stimulate fundamental research and drug discoveries. This review extensively summarizes the role of AKT/mTOR pathway in proliferation, survival, invasion, angiogenesis, autophagy, metastasis and EMT in OC, and the miRNA, lncRNA and circ RNAs that regulate this pathway in OC, as well as the therapeutic effects of AKT/mTOR inhibitors in OC. It would be a nice addition of this field. I have the following suggestions to improve the readability of this review:
- In the 2nd section, I would prefer a more concise and clearer description of the AKT/mTOR signaling pathway. It might be easier to start with a narrative explaining what the essential components of the pathway are, and then talk about mis-regulation of what components lead to tumor progression. A lot of information was included in the section, but it’s not clear to me what the relevance is in the context of tumor progression or OC. For example, I don’t see the point of mentioning the expression patterns of PI3K subclasses and different AKT isoforms.
- It is nice that Figure 1 presents an extensive summary of different aspects of the AKT/mTOR signaling pathway. However, I think a better connection between Figure 1 and the main context is needed.
- The organization of the section “Therapeutic effect of AKT/mTOR inhibitors in OC” would be much clearer if subtitles were added. Also, the second paragraph mainly talks about AKT/mTOR inhibitors in clinical trials and gives a few examples, but the logic of listing the examples is not clear. It might be better if a certain order was followed. The order could be based on monotherapy vs. combinatory usage of drugs, or effectiveness, or whether there are any side effects or not.
Author Response
Dear Sir/Ma'am,
We are grateful to you for your insightful comments. These comments are very useful for strengthening the readability and increasing the significance of the manuscript. We have now modified the manuscript as per the comments. A point by point response to the comments is mentioned below.
Comment 1:
In the 2nd section, I would prefer a more concise and clearer description of the AKT/mTOR signaling pathway. It might be easier to start with a narrative explaining what the essential components of the pathway are, and then talk about mis-regulation of what components lead to tumor progression. A lot of information was included in the section, but it’s not clear to me what the relevance is in the context of tumor progression or OC. For example, I don’t see the point of mentioning the expression patterns of PI3K subclasses and different AKT isoforms.
Response: The 2nd section “AKT/mTOR signaling pathway” has been modified. The modified section firstly states the chief elements of the AKT/mTOR pathway followed by description of the mechanistic action of each element sequentially. Also, unnecessary description of the isoforms have been removed. Only few details of the isoforms has still been included in order to link them with the following manuscript.
Comment 2:
It is nice that Figure 1 presents an extensive summary of different aspects of the AKT/mTOR signaling pathway. However, I think a better connection between Figure 1 and the main context is needed.
Response: The description of the AKT/mTOR signaling pathway in the 2nd section has been modified in accordance with the figure, describing how each subunits of the pathway leads to specific cellular processes as shown in the figure.
Comment 3:
The organization of the section “Therapeutic effect of AKT/mTOR inhibitors in OC” would be much clearer if subtitles were added. Also, the second paragraph mainly talks about AKT/mTOR inhibitors in clinical trials and gives a few examples, but the logic of listing the examples is not clear. It might be better if a certain order was followed. The order could be based on monotherapy vs. combinatory usage of drugs, or effectiveness, or whether there are any side effects or not.
Response: The section “Therapeutic effect of AKT/mTOR inhibitors in OC” has been rewritten with new subtitles and the corresponding Table 3 has also been modified accordingly. Please refer to the Table 3 attached with this mail. In the text, the theraupetic inhibitors have been first classified on the basis of preclinical and clinical studies. Subsequently, under ‘Inhibitors in clinical studies’, each inhibitor is described separately and clinical trials using these inhibitors have been discussed.
Also, please find the attachment